# Enhancing Microdroplet Image Analysis with Deep Learning

**DOI:** 10.3390/mi14101964

**Published:** 2023-10-22

**Authors:** Sofia H. Gelado, César Quilodrán-Casas, Loïc Chagot

**Affiliations:** 1Department of Computing, Imperial College London, London SW7 2AZ, UK; 2Data Science Institute, Imperial College London, London SW7 2AZ, UK; 3Department of Earth Science and Engineering, Imperial College London, London SW7 2AZ, UK; 4ThAMeS Multiphase, University College London, London WC1E 6BT, UK

**Keywords:** microdroplets, microfluidics, deep learning, computer vision, image processing

## Abstract

Microfluidics is a highly interdisciplinary field where the integration of deep-learning models has the potential to streamline processes and increase precision and reliability. This study investigates the use of deep-learning methods for the accurate detection and measurement of droplet diameters and the image restoration of low-resolution images. This study demonstrates that the Segment Anything Model (SAM) provides superior detection and reduced droplet diameter error measurement compared to the Circular Hough Transform, which is widely implemented and used in microfluidic imaging. SAM droplet detections prove to be more robust to image quality and microfluidic images with low contrast between the fluid phases. In addition, this work proves that a deep-learning super-resolution network MSRN-BAM can be trained on a dataset comprising of droplets in a flow-focusing microchannel to super-resolve images for scales ×2, ×4, ×6, ×8. Super-resolved images obtain comparable detection and segmentation results to those obtained using high-resolution images. Finally, the potential of deep learning in other computer vision tasks, such as denoising for microfluidic imaging, is shown. The results show that a DnCNN model can denoise effectively microfluidic images with additive Gaussian noise up to σ = 4. This study highlights the potential of employing deep-learning methods for the analysis of microfluidic images.

## 1. Introduction

Droplet-based microfluidics is an emerging interdisciplinary field with great potential across a wide range of scientific disciplines. It has shown great promise in biomedical applications by enabling targeted drug delivery and portable point-of-care diagnostics [1]. It is also widely used in microreactors [2] within biochemistry as well as in industrial applications such as inkjet printing [3]. Microfluidics allows easy and accurate manipulation of complex systems such as genes, molecules, and cells even for high flow rates [4], which shows great potential for the development of new technologies.

In labs, even if many techniques have been developed [5], bright-field imaging remains the most common method of monitoring droplet generation in microfluidic devices [6]. Due to the ultra-fast process of microdroplet formation, it becomes imperative to employ high-speed imaging to visualize them. However, high-speed cameras often have limited resolution or contrast, which can lead to low-resolution or noisy images and can be an issue in defining the droplet interface. Moreover, classic droplet identification methods such as manual measurements or the popular Circle Hough Transform (CHT), massively used in popular toolboxes of image processing (MATLAB 2023b, Python 3.7, or ImageJ 1.54f), need a distinct contrast at the droplet interface to show good performances. This forces the users to mismatch the refractive indices of the inner and outer phases of the droplet to thicken the interface by creating a shadow. However, it comes with a loss of precision in the exact interface location, adding a new error source in the droplet size estimation.

Therefore, there is a need for novel techniques to decrease the errors caused by the physical limitations of these microfluidic setups. Deep learning presents a promising solution to overcome these challenges by improving droplet detection and/or image quality when the resolution is too low or noisy.

For example, the open-source SAM (Segment Anything Model) released in 2023 by Meta AI can be a good alternative to CHT for droplet detection. Indeed, trained on an extensive dataset comprising over 1 billion masks from 11 million images, SAM is built on transformer-based vision models with zero-shot generalization and, therefore, does not require additional training [7]. SAM redefines image segmentation potential by achieving remarkable generalization, in contrast to single-field-focused methods. Following the recent release of SAM, various other fields promptly acknowledged its potential and began assessing the model’s suitability for their own image datasets. Notably, the medical field stands out as a prime example, with multiple studies already demonstrating the value of SAM in enhancing image segmentation tasks [8,9]. SAM’s automatic recognition of objects in complex scenarios holds promise for microfluidics applications.

Another solution is to use single-image super-resolution (SISR) methodologies to help with the lack of spatial resolution by restoring high-resolution images (HR) images from low-resolution (LR) inputs and giving access to a precise identification of tenuous details [6] (e.g., interfaces). It offers a cost-effective solution to reconstruct the visual quality of images by improving computer vision tasks such as segmentation [10] and object detection [11] without upgrading the hardware for the data collection. SISR can be represented by the following degradation model:(1)$$I_{LR}=\left(I_{HR}⊗K\right)↓S+N,$$
where $I_{HR}$ is the unknown HR image, $I_{LR}$ is the LR image, ⊗ is the convolution operator, *K* is the blurry kernel, and ↓ is the downsampling operator with a scale factor of *S* and *N* is a noise term [12]. SISR methods can be classified into three primary approaches: interpolation-based, reconstruction-based, and learning-based techniques [12]. By its inherent nature, SISR belongs to the category of ill-posed problems, which implies there exists no single solution but rather multiple potential solutions, and its performance depends on the quality of the input data. In this context, deep-learning-based methodologies demonstrate the ability to adapt and learn from diverse data and can be effectively trained to learn from the characteristics of images across varying domains, tailoring their performance to the unique characteristics of each domain [12].

Although deep-learning methods for SISR are vast, two main types of networks are commonly used: convolutional neural networks (CNNs) and generative adversarial networks (GANs).

The Super-Resolution Convolutional Neural Network (SRCNN) [13] was one of the first benchmark SISR learning methods. It was characterized by its simplicity (being a relatively shallow network) and low computational requirements. The SRCNN proved successful and has been extensively used over the years in a wide range of fields and applications, including in medicine [14] and remote sensing [15]. With the improvement of GPU capabilities and hardware systems, deeper convolutional networks have gained much attention. In this sense, the Multiscale Residual Network (MSRN) [16] employs convolution kernels of different sizes to leverage image features at different scales in the recovery of HR images. MSRN uses residual learning, which employs connections between the outputs of different layers to reduce the vanishing gradient problems of very deep network structures and lower the computational complexity of the model [16].

In recent years, GANs [17] have gained much popularity. GANs mainly work with two components: a generator trying to create fake data and a discriminator telling apart fake from real data. But even though the Super-Resolution Generative Adversarial Network (SRGAN) has shown good performance in SISR [18], GANs are notoriously known to be difficult to train and suffer from difficulty converging and instability [19].

In addition, the promising outcome in SISR can come from the Balanced Attention Mechanism (BAM) [20]. Indeed, BAM is based on a deep-learning paradigm. Attention, which is much like human vision, allows a model to focus on certain features of images during training, reducing the complexity and speeding up training. BAM resolves one of the challenges of SISR, where noise suppression by the SISR networks often leads to loss of textural information. Wang et al. [20] highlighted that the use of BAM in several scenarios, including MSRN, can improve the performance of SISR.

Finally, noise is a recurrent issue in scientific imaging, and it poses a notable obstacle to successful SISR techniques. Employing image-denoising techniques holds the potential to enable the use of images corrupted with noise while ensuring precise droplet diameter measurement. This could potentially decrease the reliance on high-quality cameras and optimal imaging conditions when capturing images. Image-denoising techniques have been extensively studied in the field of deep learning to restore clarity in noisy image data sets [21]. For example, DnCNN (denoising convolution neural network) [22] adopts a residual learning framework whereby the model learns the mapping from the noisy image to the noise in the image, the predicted noise in the image can then be removed to produce the clean image. Another notable denoising technique is the fast and flexible image-denoising network (FFDNet), which reduced the computational complexity of DnCNN and increased the flexibility of the model to different noise levels [23]. However, DnCNN has been seen to have improved performance compared to FFDNet in white image-denoising tasks such as Gaussian denoising at low noise levels (σ≤ 15), closer the noise levels present in most of the experimental microfluidic images [24].

The main objectives of this paper are to assess the performance of deep-learning methods for the accurate detection and measurement of droplets and the image restoration of LR microfluidic images. This study investigates the use of SAM as an alternative to CHT for the detection and accurate measurement of droplet diameter in images taken in a flow-focusing microchannel. In addition, the applications of three SISR methods for microfluidic images are compared. The three methods are the most prevalent interpolation technique, namely bicubic interpolation, a relatively simple learning-based method, SRCNN, and a more complex learning-based method, MSRN, with a Balanced Attention Mechanism. The study focuses on evaluating the adaptability and effectiveness of SISR in enhancing the resolution of microfluidic images. Finally, the potential of a denoising deep-learning model, DnCNN, is evaluated in denoising microfluidic images.

## 2. Materials and Methods

### 2.1. Droplet Generation and Image Acquisition

A large dataset of 20,262 images were collected at the ThAMeS-microfluidics laboratory at University College London (UCL). The experiments focused exclusively on the dripping regime and were performed in a glass flow-focusing microchannel (Dolomite Microfluidics) [25,26,27,28]. The main channel has an oval cross-section, and the dimensions are 390 µm × 190 µm (width × depth).

The side channels were filled with silicone oil (viscosity: $\mu_{c}=4.6$ mPa and density: $\rho_{c}=920$ kg m^−3^ at 20 °C) while the central channel was filled with a mixture of 52% *w/w* glycerol and 48% *w/w* water (viscosity: $\mu_{c}=6.8$ mPa and density: $\rho_{c}=1132$ kg m^−3^ at 20 °C). To avoid optical distortion and minimize the shadow effect at the droplet interface, the fluids were selected to have the same refractive index $n_{i}$ (here $n_{i}=1.39$ at 20 °C). KDS Scientific syringe pumps were used to control accurately the flow rates of continuous and disperse phases ($Q_{c}$ and $Q_{d}$), and allowed to diversify the dataset acting on the droplet size (*d*) using different flow rate combinations with $Q_{c}\in \left[0.10\;\mathrm{mL}\;\min^{-1},0.35\;\mathrm{mL}\;\min^{-1}\right]$ and $Q_{d}\in \left[0.05\;\mathrm{mL}\;\min^{-1},0.10\;\mathrm{mL}\;\min^{-1}\right]$ (see Table 1).

All images were taken with a Phantom VEO 1310 high-speed camera (12-bit depth and 1280×800 pixels resolution) equipped with a Nivatar 24× lens using an acquisition frequency of 1000 Hz and an exposure time of 10 µm. An LED backlight was used to illuminate the microchannel homogeneously. To focus on the droplet generated in the main channel, the study targeted a zone of interest of 928×288 pixels just after the microchannel inlet (see Figure 1).

### 2.2. Dataset Preparation

A random sample of 10,000 images from the collected dataset was selected and split into training, validation, and test data sets using a split of 80:10:10. Images within the selected dataset were cropped to focus on the zone of interest.

In this work, supervised deep-learning methods are used to reconstruct a super-resolved image ($I_{SR}$) from a low-resolution image ($I_{LR}$). Deep-learning models require paired input and corresponding expected output for effective training. As such, the inclusion of both HR and LR image pairs becomes imperative, allowing the model to learn the LR-to-HR mapping. During the image acquisition phase, HR images were gathered, subsequently serving as the basis for generating matching LR pairs. Bicubic interpolation was employed to produce LR counterparts across varying downsampling scales (×2, ×4, ×6, ×8). A comparison of the scale size of the HR image and the downsampled images can be seen in Figure 2.

### 2.3. Evaluation Metrics

#### 2.3.1. Droplet Segmentation and Diameter Measurement

For droplet segmentation, the Dice Similarity Coefficient (Dice) and the Jaccard Index (IoU) are employed. Both Dice and IoU are standard metrics for segmentation tasks in computer vision [29] and are defined as:(2)$$\mathrm{Dice}\left(S_{HR},S_{SR}\right)=\frac{2|S_{HR}\cap S_{SR}|}{|S_{HR}\left|+\right|S_{SR}|}$$
and
(3)$$\mathrm{IoU}\left(S_{HR},S_{SR}\right)=\frac{|S_{HR}\cap S_{SR}|}{|S_{HR}\cup S_{SR}|},$$
where $S_{HR}$ is the segmentation mask produced in the high-resolution image and $S_{SR}$ is the segmentation mask produced in the super-resolved (i.e., predicted) images.

To evaluate the detection capabilities of the droplet detection and diameter measurement methods, the percentage of droplet detection achieved in the super-resolved images is compared to the detection achieved in the HR images. Therefore, percentage detection is calculated per image by dividing the number of droplets detected in the $I_{SR}$ by the number of droplets detected in the $I_{HR}$.

In addition, since the main focus is to produce an accurate measurement of droplet diameter *d*, the absolute error (=$|d_{HR}-d_{SR}|$) and the relative error (=$|d_{HR}-d_{SR}\left|/\right|d_{HR}|$) are computed from $I_{HR}$ and $I_{SR}$. To take into account misses in the detection of droplets, two absolute errors are presented in Section 3. The Droplets Absolute Error takes into account droplets that were not detected (which will have a diameter of 0 µm). The Detections Absolute Error only takes into account droplets that were detected in both the $I_{HR}$ and $I_{SR}$. The relative error is only calculated based on the detections present in both the $I_{HR}$ and $I_{SR}$.

#### 2.3.2. Image Quality

Peak signal-to-noise ratio (PSNR) is the most commonly used metric to evaluate image restoration. PSNR has been used widely as it is simple to calculate; mathematically, it can be used for optimization, and its physical meaning is interpretable [30]. Given a high-resolution image $I_{HR}$ and a super-resolved $I_{SR}$ both with height (*H*), width (*W*), and channel (*C*) the PSNR can be defined by:(4)$$\mathrm{PSNR}=10\cdot \log_{10}\left(\frac{L^{2}}{\mathrm{MSE}}\right),$$
with
(5)$$\mathrm{MSE}=\frac{1}{HWC}{∥X-\hat{X}∥}_{2}^{2},$$
where L is the maximum pixel value, $X\in I_{HR}$ and $\hat{X}\in I_{SR}$ [6].

As can be seen from Equations (Equation 4) and (Equation 5), PSNR depends on the pixel magnitude difference between corresponding X and $\hat{X}$. Hence, a lower PSNR value does not necessarily indicate worse perceptual image quality compared to a higher PSNR value. This limitation has led to substantial criticism regarding the accuracy of PSNR as an evaluation metric in SISR [6].

Structure similarity index (SSIM) was created as an alternative image quality metric to PSNR and MSE. The SSIM takes into account the properties of the human visual system and is based on the assumption that the human visual system is naturally inclined to extract structural details from an image. The SSIM is based on the comparison of three measurements: luminance *l*, contrast *c*, and structure *s* [30]. The importance of these three components can be adjusted accordingly:(6)$$\mathrm{SSIM}={\left[l\left(X,\hat{X}\right)\right]}^{\alpha }\cdot {\left[c\left(X,\hat{X}\right)\right]}^{\beta }\cdot {\left[s\left(X,\hat{X}\right)\right]}^{\gamma },$$

A simplified version where all the components have the same importance is given by:(7)$$\mathrm{SSIM}=\frac{\left(2\mu_{X}\mu_{\hat{X}}+C_{1}\right)+\left(2\sigma_{X\hat{X}}+C_{2}\right)}{\left(\mu_{X}^{2}+\mu_{\hat{X}}^{2}+C_{1}\right)\left(\sigma_{X}^{2}+\sigma_{\hat{X}}^{2}+C_{2}\right)},$$
where $C_{1}$ and $C_{2}$ are constants, μ and σ are the mean and standard deviation of X and $\hat{X}$, respectively, and $\sigma_{X\hat{X}}$ is the covariance between X and $\hat{X}$. SSIM has been shown to provide a more accurate representation of visual quality compared to PSNR [30]. In this work, PSNR and SSIM are used together to assess image quality (see Section 3). It should be noted that SSIM and PSNR by themselves may not always be accurate indicators of visual resolution, and human perception can also help assess image quality.

### 2.4. Microdroplet Detection

Circle Hough Transform (CHT) is a well-known and commonly used method for circle detection [31,32]. It employs the Canny edge detector, which identifies the edges within an image. Following the edge detection, the equation of a circle is used to perform voting for possible positions of the circle given a range of possible radii. From the outcomes of the voting, it is possible to identify the most likely circles in the image [33]. In this study, Canny edge detection with σ=1 for noise removal before edge detection and the possible diameter of the circles detected is limited to 130 µm up to 390 µm (the depth of the channel).

The performance of CHT is compared to an adapted version of SAM. SAM performs pixel-wise segmentation masks and thus has no prior knowledge that the object of focus of the segmentation is a droplet. Since the dataset collected in this study is of the dripping regime where spherical droplets are formed, the segmentation of SAM is guided to extract the most likely circle diameter in the segmentation. Thus, SAM is employed with an additional CHT post-processing of the SAM segmentation masks and can be seen in Figure 3. In this case, since the segmentation masks can produce clear edges, CHT is used to extract the most likely droplet diameter of the detected droplets.

The performance of CHT and SAM+CHT is compared on super-resolved images using bicubic interpolation, the most common and simple method of SISR studied in this work for scales ×2, ×4, ×6 and ×8. To compare the performance of CHT and SAM+CHT, the segmentation performance is assessed using Dice and IoU, the error in the diameter measurement using absolute errors and relative errors, and the percentage of droplets detected. The best droplet detection and diameter measurement method is identified and is employed in the rest of the study.

### 2.5. Deep-Learning Super-Resolution Models

In this section, two learning-based super-resolution models are studied: an SRCNN model and an MSRN model enhanced with a Balanced Attention Mechanism (BAM). In this work, the image restoration and segmentation performances are compared using both learning-based models against the baseline: super-resolved images using bicubic interpolation. The architecture and implementation of both learning-based super-resolution models are described and compared below.

#### 2.5.1. SRCNN

Initially, a simple SRCNN was implemented for the SISR task. Following the original implementation by [13], the SRCNN model created was a three-layer network with filter sizes: (64 × 1 × 9 × 9), (32 × 64 × 5 × 5), (1 × 32 × 5 × 5). The input images to the network are upscaled LR images using bicubic interpolation. The SRCNN aims to perform an end-to-end mapping of $I^{LR}$ to $I^{SR}$. Thus, a different model needs to be trained for every upscaling factor (×2, ×4, ×6, ×8).

The network performs three main operations. First, patches from the image are extracted and represented as a feature map. Next, non-linear mapping is performed by mapping each of the patches into a lower-dimensional space. Finally, reconstruction of the HR image is achieved through a final convolutional layer [13].

The SRCNN model was trained for 300 epochs, a batch size of 32 images with MSE loss, and using Adam optimizer. MSE loss is employed as it has been seen to favor a high PSNR. The best-performing set of model parameters was selected according to the highest PSNR achieved on the validation dataset. The model was then evaluated on the unseen test dataset using image quality metrics (PSNR and SSIM) and the droplet detection and segmentation metrics outlined in Section 2.3.1.

#### 2.5.2. MSRN-BAM

The MSRN model is a benchmark in image super-resolution. To improve the performance of MSRN, a Balanced Attention Mechanism (BAM) was incorporated into the model. BAM is composed of two attention mechanisms: ACAM and MSAM. The ACAM module is responsible for suppressing noise in the upsampled feature maps, and MSAM attention is focused on capturing high-frequency texture details. The parallelization of ACAM and MSAM allows the BAM module to self-optimize during the backpropagation process to achieve a balance between noise suppression and texture restoration [20].

The model architecture comprises two main segments: the feature extraction module and the image reconstruction module. The feature extraction module incorporates the multiscale residual block (MSRB) and hierarchical feature fusion structure (HFFS). Figure 4 shows the architecture of the MSRN model with the BAM employed.

The MSDRB-BAM is designed to detect image features across various scales, consisting of multiscale feature fusion and local residual learning components. Each MSDRB-BAM module consists of 2 sets of convolutional layers with ReLU activation and kernel sizes (3 × 3 and 5 × 5). Between the two sets of layers, the intermediate outputs are concatenated. This is followed by the BAM layer for improved feature selection. Finally, the MSRN-BAM uses a residual connection that combines the attention-modulated output with the original input. The overall architecture of the MSRN-BAM has 8 blocks of MSDRB-BAM.

The HFFS then takes the feature outputs from each MSRB and effectively fuses them as the network progresses toward generating the final output. This fusion occurs in the tail of the network architecture, where the concatenated features from MSDRB-BAM blocks undergo transformations through convolutional layers, activations, and upsampling. The goal of HFFS is to blend information from various layers together so that important details are not lost, resulting in a precise image reconstruction. This strategy addresses the difficulty of maintaining useful information throughout deep networks, resulting in a high-quality result.

The training data are enhanced through random cropping, random horizontal and vertical flips, and random 90-degree rotations. Each training batch consists of 16 randomly extracted LR patches with size 32 × 32. The model is trained for 400 epochs. The model undergoes training using the Adam optimizer with the learning rate of 1 × 10^−4^ and employs L1 as the loss function.

In comparison with SRCNN, where the LR images are upsampled to the dimension of the HR image via bicubic interpolation, MSRN-BAM uses an unamplified LR image as the input of the network, which is upsampled to the HR dimensions by the network. In addition, MSRN-BAM is a considerably deeper network with separate modules to combine multiscale information and capture the most relevant features of the super-resolution task.

### 2.6. Image Denoising with DnCNN

The DnCNN is a deep neural network designed for image denoising. The input to the DnCNN is a noisy image, which can be represented by $I_{noisy}=I_{clean}+N$. DnCNN employs a residual approach where $R(I_{noisy})\approx N$, and as such, the clean image can be obtained from $I_{clean}=I_{noisy}-R\left(I_{noisy}\right)$. Therefore, the model tries to learn the mapping from the noisy image to the noise present in the image, and the loss function calculates the difference between the predicted noise in the input image $R(I_{noisy})$ and the actual noise present in the input image $R\left(I_{noisy}\right)-I_{clean}$. The DnCNN architecture can be seen in Figure 5.

The DnCNN consists of three main types of layers. The first layer is a Conv+ReLU layer using a kernel size of 3 to generate feature maps. The second type of layer is a Conv+BN+ReLU layer, which forms the main body of the network. A total of 17 of these layers are employed. The final layer is a Conv layer, which reconstructs the residual image. The model follows residual learning where the predicted residual image corresponds to the noise present in the noisy image. Thus, the final clean output image is generated by subtracting the residual image from the input image.

To train and test the DnCNN, the dataset is corrupted with additive Gaussian noise. DnCNN networks are trained using a range of Gaussian noise levels (σ=2, σ=3, σ=4, σ=6). The DnCNN is trained using patches of size 40 × 40. The batch size employed is of size 32 and a learning rate of 1 × 10^−4^ was employed. DnCNN uses Adam optimizer and L1 loss, similar to MSRN-BAM. Throughout the training process, the model is evaluated on the validation set, and the model is trained for 1200 epochs. The performance of the model is then evaluated on the unseen test dataset. The improvement in image quality and the performance of the denoised dataset using MSRN-BAM and SAM+CHT is compared to the clean dataset.

## 3. Results

### 3.1. Droplet Detection

In this section, the droplet detection and segmentation performance of CHT and SAM+CHT are compared across the scale of super-resolved images using bicubic interpolation. Table 2 shows that the performance of detection and segmentation is superior for SAM+CHT compared to CHT for all scales. With increasing scale, the performance of CHT deteriorates rapidly. The results show that using CHT for droplet detection will produce large absolute errors in the detection and will struggle to detect droplets when the images are beyond an ×2 scale using bicubic interpolation (e.g., 39% detection for ×4). This is because, as part of CHT, the Canny edge detector is used to detect edges. When these edges are too fine, as is the case with the droplets of this study, it is difficult for the next stage (voting stage) of the CHT to identify the curves in the image and thus accurately predict the position and the diameter of the droplet. From ×4, CHT has a poor Dice and IoU performance with increasing scale. SAM+CHT performs consistently across almost all scales with high Dice/IoU, percentage detection, and low absolute and relative errors in the detected droplets. For ×4, CHT shows Dice and IoU of, respectively, 49% and 37% with a 39% droplet detection, while SAM+CHT has a droplet detection of 92% with Dice = 94% and IoU = 90%. It is only for ×8 that SAM+CHT shows a drop in performance with 57% droplet detection and Dice = 68% and IoU = 55%.

An example of the detections produced by CHT and SAM+CHT with a bicubic interpolated image at the ×4 scale can be seen in Figure 6. Figure 6 shows that CHT struggles to detect the droplets even in the HR image in comparison to SAM+CHT, which successfully detects and segments all the droplets in the image. Although SAM+CHT misses a droplet in the bicubic image, its performance is considerably superior to CHT.

### 3.2. Super-Resolution Models

The SRCNN and MSRN-BAM models were trained as specified in Section 2.5. The performance of the models was evaluated on an unseen test dataset of 1000 images. To measure the super-resolution capabilities of the models, the PSNR and SSIM of the super-resolved images with respect to the HR images were calculated and are shown in Table 3. The performance of the SRCNN and MSRN-BAM is compared to standard bicubic interpolation. Based on the PSNR and SSIM, both SRCNN and MSRN-BAM produce slightly higher-quality images, based on the PSNR and SSIM, compared to the standard bicubic interpolation. Out of the two learning-based methods, Table 3 shows systematically that MSRN-BAM outperforms SRCNN at every scale according to the image quality metrics.

Figure 7 shows the improvement of PSNR and SSIM for the MSRN-BAM network during training for the different scales. Notably, the PSNR computed on the validation dataset fluctuates during training, reaching a more stable behavior after epoch 310. The fluctuations in the PSNR during training observed in Figure 7 are common in neural network training. As such, drops in the PSNR during training, such as in epoch 80 for ×4 scale and epoch 190 in ×8, could be attributed to an update in model parameters that deviated from the optimal configuration, leading to a temporary decline in performance. This unstable behavior during training can be attributed to several factors, such as the stochastic nature of gradient descent. The model training was stopped at 400 epochs since the PSNR did not improve by more than 0.03 in the last 50 epochs.

Segmentation performance for the learning methods and bicubic interpolation are also assessed using the SAM+CHT segmentation model outlined in Section 2.5. As shown in Table 2, SAM+CHT can cope with high levels of image degradation where high Dice/IoU and Percentage detections can be obtained in ×2 and ×4 scale. Notably, the absolute error for the detected droplets is consistent and low across all scales, showing that the droplets detected are accurate to detections in the HR images. However, even if SRCNN and the bicubic interpolation method show similar PSNR and SSIM than MSRN-BAM, a degradation of the performance can be observed in downsampling scales (×6 and ×8) showing the production of lower quality images for both methods. Indeed, for SRCNN and the bicubic interpolation method, the detection drops below 80% for ×6 and their IoU below 60% for ×8. On its side, MSRN-BAM gives constant high performance from ×2 to ×8 with an IoU ~90% and a droplet detection >90% which confirms the higher image quality produced already suggested by the PSNR and the SSIM.

Figure 8 shows that MSRN-BAM super-resolved images SAM+CHT achieves excellent detection and low absolute error for droplets of size < 250µm at all scales. Although percentage detections >80% can still be achieved for scales ×2 and ×4 for droplets of size > 250µm, the performance worsens for scales ×6 and ×8 for these droplet sizes. This drop in performance could be attributed to the challenge of discerning the droplet interface from the channel wall when the droplet size is similar to the channel depth (390 µm).

The performance of SAM+CHT can be compared with the work of Vo et al. [31], who employed CHT for their image-based feedback system to control droplet generation. They observed poor detection for diameter *d* > 100 pixels and no successful droplet detections for *d* > 180 pixels. Here, the super-resolved images using MSRN-BAM and SAM+CHT show that a good detection (more than 70% of droplets detected) can be achieved for all scales for droplets diameter from 96 pixels to 230 pixels (125 µm to 275 µm).

### 3.3. Image Denoising

The DnCNN network was trained and tested on downsampled images by a ×4 scale with additive Gaussian noise (2 ≤ σ ≤ 6), see Table 4. Due to the low contrast between the phases (with refractive index matching) in the dataset of microfluidic images employed in this study, only noise levels 6 ≤ σ were employed. Figure 9 shows the improvement in PSNR and SSIM during the DnCNN training on the validation dataset. Notably, while σ > 2 shows an improvement in the PSNR and SSIM during training, σ = 2 remains stable with high PSNR and SSIM. This could be attributed to the low level of noise at σ = 2. After the first optimization of the model parameters, the model may have found its optima. Table 4 shows the improvement in PSNR and SSIM of the test set for the noisy images and the denoised images after applying the DnCNN model. Notably, the standard deviation of the PSNR increases with increasing noise levels, showing that for the DnCNN model, a wider range of image quality in the predicted clean images is produced at higher noise levels.

Following the denoising of the test images, the super-resolution model MSRN-BAM and segmentation using SAM+CHT was applied to test the segmentation performance on denoised images (see Table 5). High Dice and percentage detections were achieved until σ≤4. The detection performance decreases notably for σ = 6. However, the absolute error of the detected images remained low (<6 µm), showing that detected droplets were accurate to the detections made in the HR images. Figure 10 shows the improvement in image quality after the denoising model and the super-resolution model. The denoising model removes the additive Gaussian noise while preserving the structural components of the image. The super-resolution model shows an improvement in the high-frequency detail of the image, which can be appreciated from the fine droplet borders.

## 4. Discussion

CHT has been employed intensively in microfluidic applications for droplet detection [31,32]. However, the results of this study show that CHT is a poor droplet detection algorithm for microfluidic systems where the droplet interface is fine and the contrast is small due to good index matching. Moreover, the results show that CHT is challenged by low-resolution images where the droplet interface may become more pixelated and less defined, worsening the contrast between the droplet and the background.

In contrast, SAM+CHT achieves high performance across all scales, only seeing a significant drop in the segmentation Dice score and percentage detection for the ×8 scale. In addition, SAM+CHT is shown to be an accurate detection method as detections made have a small absolute and relative error.

An alternative droplet detection method suggested by Mudugamuwa et al. [34] is thresholding. In their work, they obtain RGB images and select the color band, which produces the largest contrast between the background and the droplets to use for thresholding and follow this process by morphological operations such as erosion and dilation to obtain a smooth droplet interface. However, for their thresholding system to be able to differentiate between the water droplets and the coconut oil flow, it required coloring the water with a green pigment. As with CHT, thresholding requires a high contrast between the droplets and the background. In addition, thresholding needs post-processing morphological operations, which may modify the shape of the droplet interface.

Thus, this work shows that SAM+CHT is a robust and accurate detection and diameter measurement tool for droplets in a microfluidic system. SAM+CHT can be used to detect droplets with fine interfaces without needing to change the properties of the water or oil phases.

For low-resolution images, the potential of learning methods was proved compared to the classic bicubic interpolation method. However, MSRN-BAM shows a better performance compared to SRCNN, which can be attributed to several factors. First, SRCNN uses the upsampled bicubic interpolated image as the input to the network. In comparison, MSRN-BAM learns the upsampling operation in the feature space. Bicubic interpolation introduces smoothing effects, which may result in poor predictions of the image composition in the SRCNN model [12]. In addition, SRCNN is a relatively shallow network compared to MSRN-BAM. Shallow networks tend only to capture low-level features (contour, edges, angles) and miss high-frequency features [35]. In the case of the microfluidic images used in this study, the edges of the microdroplets were very fine and became more distorted with increasing scale. As the input to the SRCNN was the bicubic interpolated image, the loss in the definition of the edges of the droplet may have made it challenging for the network to produce high-resolution images with increasing scale. Furthermore, the SRCNN network uses MSE loss in comparison with the L1 loss employed by MSRN-BAM. MSE loss has been extensively used since it is known to favor high PSNR; however, Zhao et al. [36] showed networks trained with L1 loss achieved improved image restoration compared to those trained with MSE loss, which produced smoother textures. As can be seen in Figure 7, the L1 loss is able to guide the model optimization towards improved PSNR and SSIM during the training on the validation dataset. MSRN-BAM proves to be an effective SISR method for resolving images of up to an ×8 scale to a high standard of quality. Together with SAM+HCT, the results show it is possible to super-resolve and detect droplets with a high level of accuracy and detection.

Similarly, Rutkowski et al. [37] employed YOLO, a well-known object detection algorithm, to detect and measure droplets in a range of microfluidic experiments. The authors test YOLO on a range of experiments using a water-continuous phase with varying contrast of the dispersed oil phase. Their analysis shows YOLO provides improved detection, especially in low-contrast media, to those obtained using CHT methods applied in ImageJ. Nevertheless, the authors do not test the robustness of YOLO to image resolution, and their weakest contrast between the phases was still greater than the one proposed in this study. In addition, YOLO requires training on annotated droplet images, a process that demands substantial human effort for annotation. In comparison, SAM+CHT requires no previous training.

Finally, denoising models can be used to restore the quality of blurry or spatially unresolved images. DnCNN model combined with MSRN-BAM applied to ×4 downsampled images showed remarkable performances for noise levels of σ≤4 using a Gaussian noise. The low Dice score, absolute error, and percentage may be due to the loss of the structural features of the image during the denoising process in the presence of large amounts of Gaussian noise.

In other vision-based droplet detection systems, little notice has been given to the effect of noise on droplet detection. Rutkowski et al. [37] employs ImageJ’s non-local means-denoising plugin (smoothing factor = 2, σ = 15); however, they do not experiment with adding noise to their images. Although inbuilt image-denoising libraries may provide acceptable image quality in images that do not contain much noise, they may alter the structure of the droplets since they have no prior knowledge of similar image transformations. In this aspect, deep-learning denoising networks such as DnCNN trained on microfluidic images may hold an advantage, allowing for larger amounts of noise in images while being able to accurately predict the denoised images without ’guessing’ the image structure but rather informed on learned transformations on similar microfluidic images.

In conclusion, this work shows the potential of deep-learning methods for accurate detection and measurement of droplets in microchannels using low-resolution images. The proposed SAM+CHT method for droplet detection and diameter measurement proves to provide an accurate measurement of droplet diameter and is more robust to image quality than the most commonly used method, CHT. SISR deep-learning models also show improvement in image quality and droplet detection and measurement using low-resolution images. Notably, the deepest network studied MSRN-BAM achieves consistent segmentation performance across all resolution scales (×2, ×4, ×6, ×8). This study finds detections for droplets within diameter 125 µm to 250 µm achieve the highest percentage detection and smallest absolute error for the dataset collected. In addition, this work shows the potential of deep learning denoising models in microfluidic imaging. DnCNN can improve image quality and obtain comparable segmentation performance to clean images for σ < 6. Lastly, this study effectively establishes the promising capabilities of deep learning in accurate droplet diameter detection and image restoration, current challenges within the context of microfluidics imaging. Indeed, for simple microfluidic systems, as shown in this study, deep learning methods are already apt to enhance image processing significantly. For low-spatial-resolution images, we recommend using MSRN-BAM to improve the image quality before the droplet detection. Moreover, if the images show a Gaussian noise below σ≤4, then DnCNN can be used along MSRN-BAM. For droplet detection, SAM+CHT should be preferred to CHT to increase the detection rate and decrease the uncertainty on the droplet diameter, especially for low-contrast images.

Although the methodologies used here aim to be extended to more complex systems, as densely packed droplets are often discussed in the literature [38], some potential limitations can be raised. For example, future work should focus on the segmentation and detection of such complex configurations where the droplets may overlap or be deformed by the confinement. One strategy could be to develop hybrid models that merge classical image processing techniques with advanced ML approaches. Given the added complexity where droplets overlap or deform due to confinement, a promising strategy might involve the utilization of a specific U-Net architecture, which can be tweaked to accommodate the distinction between closely packed droplets or discerning the boundaries of deformed droplets. Furthermore, active learning can be incorporated, enabling the model to improve its predictive performance continuously over time. With active learning, the model could identify the most informative samples and then effectively learn to recognize other complex examples with fewer annotated examples.

## Figures and Tables

**Figure 1 micromachines-14-01964-f001:**
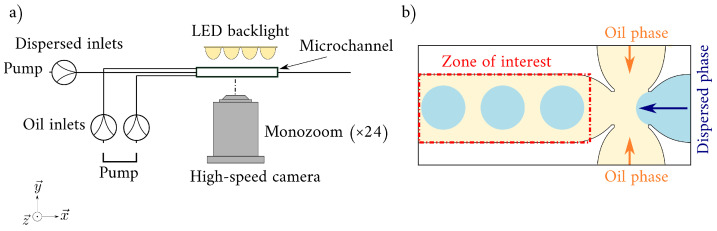
(**a**) Sketch of the experimental setup. (**b**) Sketch of the microchannel during the dripping regime. The red dashed line shows the zone of interest (928×288 pixels) used for this study.

**Figure 2 micromachines-14-01964-f002:**
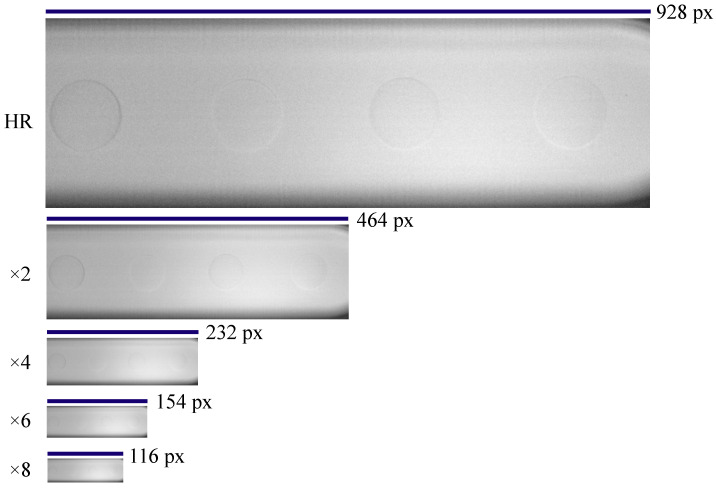
Width in pixels of high-resolution (HR) image and corresponding downscaled images at scales ×2, ×4. ×6 and ×8.

**Figure 3 micromachines-14-01964-f003:**
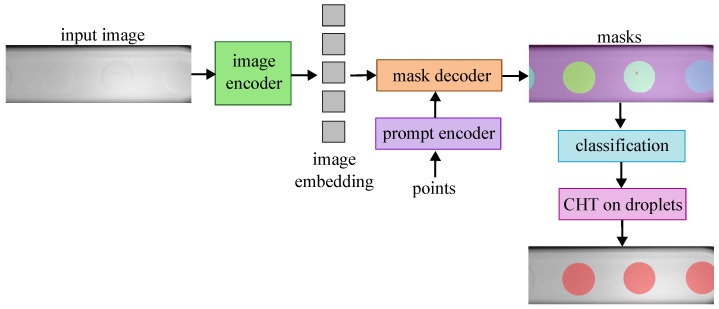
Architecture of the Segment Anything Model followed by Circular Hough Transform (SAM+CHT) for droplet detection. Automatic segmentation of the whole image was performed using evenly sampled points as the prompt for the prompt encoder.

**Figure 4 micromachines-14-01964-f004:**
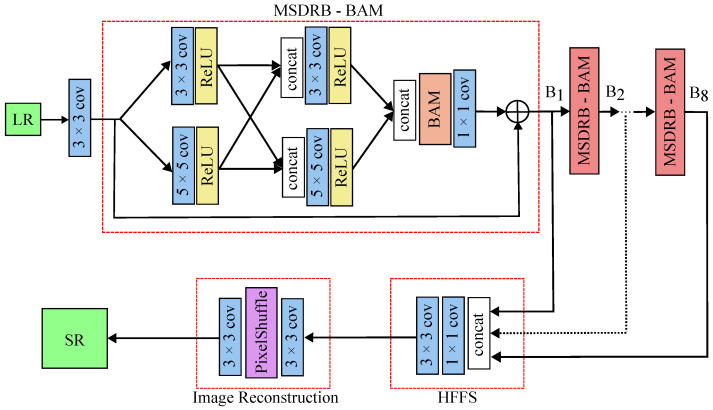
Architecture of the super-resolution model MSRN-BAM. The input of the model is the low-resolution (LR) image; layers employed include convolutional layers (conv) of kernel sizes 3 × 3 and 5 × 5, rectified linear unit (ReLU) layers, Pixel Shuffle layer and concatenation operation. The main unit of the network is the MSDRB-BAM block; the model has 8 MSDRB-BAM blocks. The output of the model is the super-resolved image (SR).

**Figure 5 micromachines-14-01964-f005:**
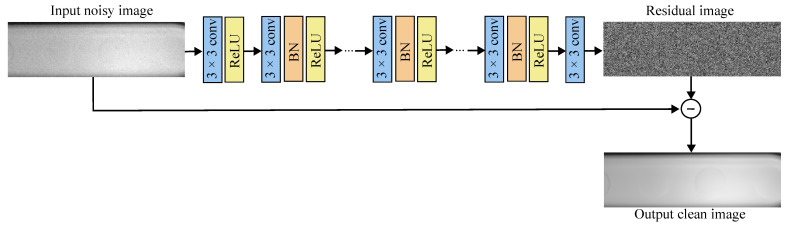
Architecture of the image-denoising DnCNN model. I The input to the model is the noisy image. The layers employed include convolutional layers (conv) of kernel sizes 3 × 3, rectified linear unit (ReLU) layers, and batch normalization (BN) layers. The main body of the model predicts the residual image (noise present in the noisy image). The final output of the model is the clean denoised image.

**Figure 6 micromachines-14-01964-f006:**
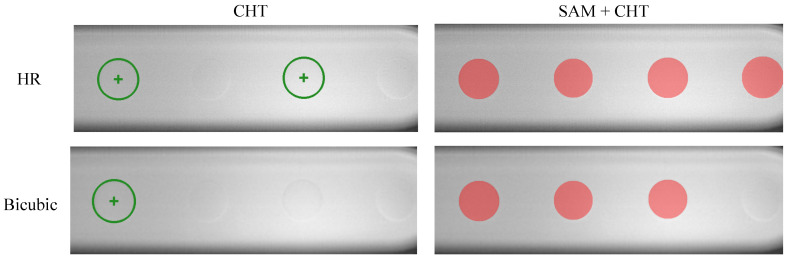
Comparison of Circular Hough Transform (CHT) and Segment Anything with Circular Hough Transform (SAM+CHT) on a high-resolution image (HR) and bicubic interpolated SISR method for ×4 scale.

**Figure 7 micromachines-14-01964-f007:**
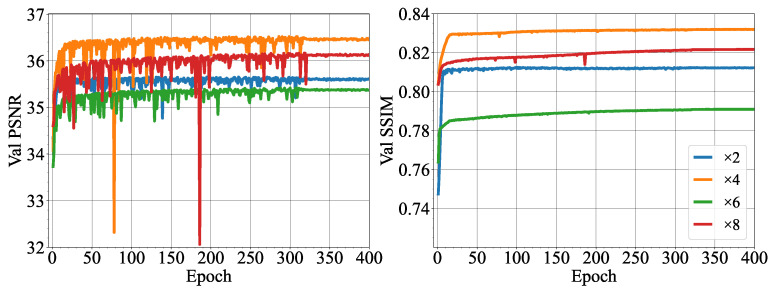
Peak signal-to-noise ratio (PSNR) and structural similarity index (SSIM) of the MSRN-BAM model on the validation dataset during training for networks trained at different super-resolution scales.

**Figure 8 micromachines-14-01964-f008:**
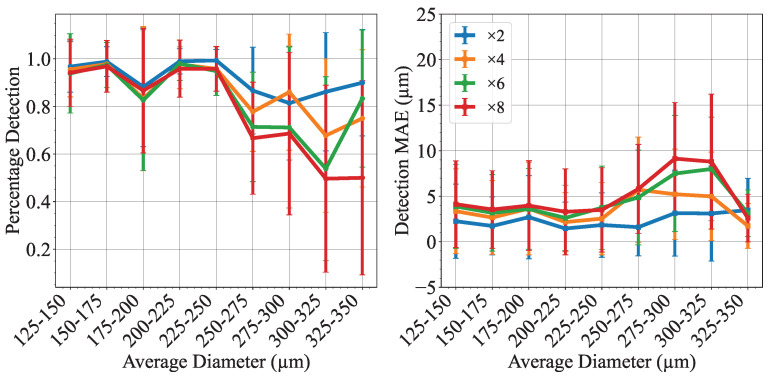
Percentage droplet detection and absolute error for the MSRN-BAM model at different resolution scales on the test dataset grouped by droplet diameter. Error bars show standard deviation.

**Figure 9 micromachines-14-01964-f009:**
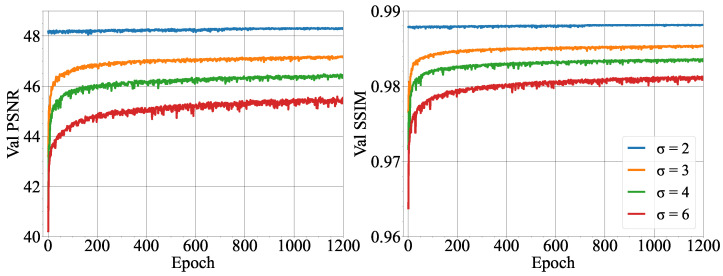
Peak signal-to-noise ratio (PSNR) and structural similarity index (SSIM) of DnCNN model on the validation dataset during training for networks trained using additive Gaussian noise with a range of noise levels.

**Figure 10 micromachines-14-01964-f010:**
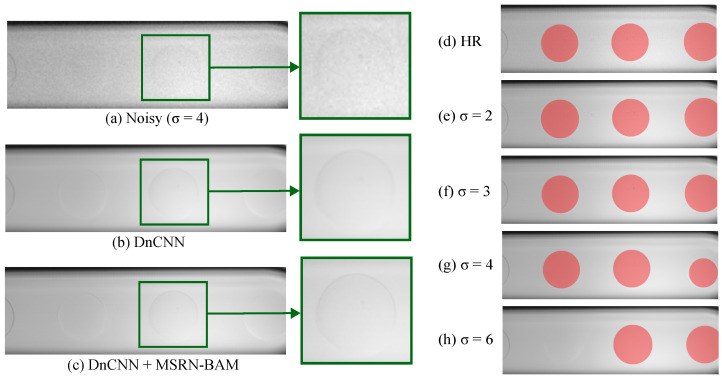
(**a**,**b**) Performance of DnCNN on denoising an image corrupted with Gaussian additive noise. (**c**) Improvement in resolution using a Multiscale Residual Network with Balanced Attention Mechanism (MSRN-BAM) at a ×4 scale. Comparison of the performance of Segment Anything with Circular Hough Transform (SAM+CHT) on (**d**) high-resolution (HR) image and (**e**–**h**) denoised super-resolved images.

**Table 1 micromachines-14-01964-t001:** Table of the 10,000 experiments randomly selected for the 20,262 initial experiments. $Q_{c}$ and $Q_{d}$ are the continuous and dispersed flow rates, and *d* is the mean droplet diameter with its standard deviation.

$Q_{c}$	$Q_{d}$	*d*	Number of Images
mL min^−1^	mL min^−1^	µm	-
0.10	0.05	211.74 ± 3.49	972
0.10	0.07	225.71 ± 2.39	967
0.10	0.10	298.71 ± 15.03	973
0.20	0.05	177.03 ± 3.18	1061
0.20	0.10	190.10 ± 4.98	1009
0.25	0.07	170.18 ± 3.55	999
0.30	0.05	153.72 ± 2.86	1020
0.30	0.07	156.58 ± 2.33	1038
0.35	0.05	145.15 ± 2.48	970
0.35	0.07	140.18 ± 2.40	991

**Table 2 micromachines-14-01964-t002:** Detection and segmentation results of Circular Hough Transform (CHT) and Segment Anything with Circular Hough Transform (SAM+CHT). The table shows the results using bicubic interpolation as the single-image super-resolution method at different scales. The absolute error is reported for all droplets (Droplets Absolute Error) and detected droplets (Detections Absolute Error and Detections Relative Error).

Method	Scale	Dice	IoU	Droplets Absolute Error (µm)	Detections Absolute Error (µm)	Detections Relative Error	Percentage Detection
CHT	×2	0.84 ± 0.19	0.77 ± 0.24	42.98 ± 77.57	2.30 ± 4.09	**0.01 ± 0.02**	0.79 ± 0.27
	×4	0.49 ± 0.30	0.37 ± 0.27	115.25 ± 89.63	6.25 ± 7.80	0.03 ± 0.03	0.39 ± 0.30
	×6	0.21 ± 0.22	0.14 ± 0.17	163.40 ± 71.21	11.27 ± 20.49	0.05 ± 0.06	0.11 ± 0.21
	×8	0.12 ± 0.13	0.07 ± 0.09	183.39 ± 49.20	95.92 ± 83.73	0.28 ± 0.24	0.01 ± 0.06
SAM+CHT	×2	**0.96 ± 0.10**	**0.93 ± 0.10**	**11.45 ± 45.76**	**2.01 ± 3.63**	**0.01 ± 0.02**	**0.95 ± 0.16**
	×4	**0.94 ± 0.10**	**0.90 ± 0.10**	**18.10 ± 56.20**	**3.70 ± 4.80**	**0.03 ± 0.02**	**0.92 ± 0.20**
	×6	**0.84 ± 0.20**	**0.77 ± 0.20**	**42.05 ± 81.40**	**4.46 ± 5.22**	**0.02 ± 0.03**	**0.78 ± 0.28**
	×8	**0.68 ± 0.20**	**0.55 ± 0.20**	**81.60 ± 97.20**	**5.39 ± 5.74**	**0.03 ± 0.03**	**0.57 ± 0.27**

Best performance indicated in bold.

**Table 3 micromachines-14-01964-t003:** Image quality metrics for three SISR methods: bicubic interpolation, super-resolution convolutional neural network (SRCNN), and Multiscale Residual Network with Balanced Attention Mechanism (MSRN-BAM). Image quality metrics include peak signal-to-noise ratio (PSNR) and structural similarity index (SSIM).

Method	Scale	PSNR	SSIM
Bicubic	×2	35.38 ± 0.10	0.75 ± 0.01
	×4	36.20 ± 0.10	0.76 ± 0.01
	×6	34.81 ± 0.20	0.70 ± 0.02
	×8	34.90 ± 0.40	0.72 ± 0.02
SRCNN	×2	35.57 ± 0.10	**0.77 ± 0.01**
	×4	36.50 ± 0.10	0.78 ± 0.02
	×6	35.24 ± 0.10	0.71 ± 0.02
	×8	35.86 ± 0.20	0.74 ± 0.02
MSRN-BAM	×2	**35.65 ± 0.10**	**0.77 ± 0.01**
	×4	**36.51 ± 0.10**	**0.79 ± 0.01**
	×6	**35.40 ± 0.10**	**0.74 ± 0.02**
	×8	**36.16 ± 0.10**	**0.77 ± 0.02**

Best performance indicated in bold.

**Table 4 micromachines-14-01964-t004:** Comparison of image quality metrics for noise corrupted images and denoised images using DnCNN for different noise levels. Image quality metrics include peak signal-to-noise ratio (PSNR) and structural similarity index (SSIM).

Sigma	PSNR Noisy	PSNR DnCNN	SSIM Noisy	SSIM DnCNN
2	41.84 ± 0.00	47.03 ± 0.08	0.93 ± 0.01	0.98 ± 0.00
3	38.52 ± 0.00	46.27 ± 0.12	0.87 ± 0.01	0.98 ± 0.00
4	36.09 ± 0.00	45.74 ± 0.16	0.79 ± 0.02	0.98 ± 0.00
6	32.94 ± 0.00	45.23 ± 0.22	0.67 ± 0.02	0.97 ± 0.00

**Table 5 micromachines-14-01964-t005:** Detection and segmentation of Segment Anything with Circular Hough Transform (SAM+CHT) on denoised images of the test set super-resolved using the Multiscale Residual Network with Balanced Attention Mechanism (MSRN-BAM) for a ×4 scale. The error is reported for all droplets (Droplets Absolute Error) and detected droplets (Detections Absolute Error and Detections Relative Error).

Method	Sigma	Dice	IoU	Droplets Absolute Error (µm)	Detections Absolute Error (µm)	Detections Relative Error	Percentage Detection
DNCNN + MSRN-BAM	Clean	0.94 ± 0.10	0.90 ± 0.10	16.90 ± 54.52	3.01 ± 4.30	0.02 ± 0.03	0.93 ± 0.19
	2	0.94 ± 0.12	0.90 ± 0.15	18.62 ± 58.52	3.35 ± 4.44	0.02 ± 0.03	0.92 ± 0.20
	3	0.91 ± 0.13	0.86 ± 0.17	29.10 ± 71.00	4.06 ± 4.94	0.02 ± 0.03	0.86 ± 0.25
	4	0.86 ± 0.17	0.78 ± 0.21	42.62 ± 80.63	4.45 ± 5.31	0.03 ± 0.03	0.78 ± 0.27
	6	0.66 ± 0.24	0.53 ± 0.24	93.79 ± 98.42	5.68 ± 5.85	0.03 ± 0.03	0.53 ± 0.26

## Data Availability

The code and models employed for this study are publicly available at: https://github.com/SoFia2401/ImageAnalysis_Microfluidics (accessed on 19 October 2023). The project repository contains the trained models from this study as well as the code to train and test the models using a new dataset.

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
