# Peer review of "Enhancing Microdroplet Image Analysis with Deep Learning"

_micromachines, 2023, doi:10.3390/mi14101964_

Round 1

Reviewer 1 Report

This research paper investigates the transformative potential of deep learning in the domain of microfluidics. It primarily focuses on two crucial aspects: accurate droplet detection and measurement, and image restoration. The study showcases the Segment Anything Model (SAM) as a superior alternative to the widely used Circular Hough transform for droplet detection, particularly excelling in low-contrast microfluidic images. Moreover, the research introduces a deep learning super resolution network, MSRN-BAM, which effectively enhances image resolution at various scales, offering results comparable to high-resolution images. Additionally, the paper explores the broader scope of deep learning in microfluidics by demonstrating its capability in tasks like image denoising using the DnCNN model. These findings underscore the immense potential of deep learning methods to revolutionize the accuracy and reliability of microfluidic image analysis.

This paper showcases a remarkable novelty in seamlessly merging the domains of deep learning and droplet processing, introducing an innovative approach with substantial application value for the entire field of microfluidics research. The meticulous experiment layout, rigorous methodology, and evident professionalism in conducting the research set a high standard for future studies in the field. The findings resonate with the ever-growing demand for more efficient and accurate microfluidic processes, making this paper a highly recommendable candidate for publication. The comprehensive insights and innovative methodologies presented in this paper are poised to make a substantial impact in advancing microfluidics research and should be shared with the scientific community. Considering its high quality, I would recommend published with minor revision.

Minor:

1.    The language use in the paper is commendable, effectively conveying the essential aspects of the study. The paper begins by succinctly introducing the research topic and its significance, which serves as an excellent hook for readers. Furthermore, the key objectives and methodology are clearly outlined, which aids in providing a quick overview of the study's approach. Overall, the language use in the paper is of high quality, effectively conveying the research findings. Only minor changes are required to further enhance sentence fluency and readability.

2.    While this paper undeniably offers an efficient solution for the precise detection of droplet sizes in microfluidic devices, it is indeed important to consider the practicality of these solutions in real-world scenarios. In many actual applications, droplets are often densely packed in confined spaces, making segmentation and detection significantly more challenging. It would be highly beneficial if the authors could provide deeper insights or address the applicability of their methodology in such complex situations. Elaborating on the potential limitations and discussing strategies or adaptions to handle scenarios with densely packed droplets would enhance the paper's practical value and its relevance to a broader range of real-world microfluidics applications. This additional layer of insight could make the paper even more impactful by addressing the complexities that researchers may encounter in their work.

3.    Figure 7 illustrates the training history of MSRN-BAM, and it would enhance the paper's clarity if the authors provided a more comprehensive explanation of the stopping criteria used during the training process. The figure suggests that the validation peak signal-to-noise ratio (Val PSNR) is still showing gradual improvement, prompting the consideration that continued training might potentially result in the development of a state-of-the-art model. A more detailed account of the rationale behind the chosen stopping point and an exploration of the possibility of further refining the model would offer readers a deeper understanding of the methodology and its potential for achieving cutting-edge performance.

4.    In Figure 7, a conspicuous feature is the abrupt decline in performance around epoch 190, particularly noticeable in the red line of the x8 experiment. Such drops are not uncommon in the realm of CNNs and could be attributed to factors like overfitting, variations in the training data, or fluctuations in learning rates. To enhance the figure's interpretability, a concise explanation by the authors regarding the cause of this dip would be valuable.

1.    The language use in the paper is commendable, effectively conveying the essential aspects of the study. The paper begins by succinctly introducing the research topic and its significance, which serves as an excellent hook for readers. Furthermore, the key objectives and methodology are clearly outlined, which aids in providing a quick overview of the study's approach. Overall, the language use in the paper is of high quality, effectively conveying the research findings. Only minor changes are required to further enhance sentence fluency and readability.

Author Response

We thank the reviewer for their careful reading of our manuscript and constructive feedback and respond below (in blue) to the queries raised.
The modifications made to the paper are highlighted in red to facilitate the review of our revised paper.

Comments to the Author

This research paper investigates the transformative potential of deep learning in the domain of microfluidics. It primarily focuses on two crucial aspects: accurate droplet detection and measurement, and image restoration. The study showcases the Segment Anything Model (SAM) as a superior alternative to the widely used Circular Hough transform for droplet detection, particularly excelling in low-contrast microfluidic images. Moreover, the research introduces a deep learning super resolution network, MSRN-BAM, which effectively enhances image resolution at various scales, offering results comparable to high-resolution images. Additionally, the paper explores the broader scope of deep learning in microfluidics by demonstrating its capability in tasks like image denoising using the DnCNN model. These findings underscore the immense potential of deep learning methods to revolutionize the accuracy and reliability of microfluidic image analysis.

This paper showcases a remarkable novelty in seamlessly merging the domains of deep learning and droplet processing, introducing an innovative approach with substantial application value for the entire field of microfluidics research. The meticulous experiment layout, rigorous methodology, and evident professionalism in conducting the research set a high standard for future studies in the field. The findings resonate with the ever-growing demand for more efficient and accurate microfluidic processes, making this paper a highly recommendable candidate for publication. The comprehensive insights and innovative methodologies presented in this paper are poised to make a substantial impact in advancing microfluidics research and should be shared with the scientific community. Considering its high quality, I would recommend published with minor revision.

Minor:

1 The language use in the paper is commendable, effectively conveying the essential aspects of the study. The paper begins by succinctly introducing the research topic and its significance, which serves as an excellent hook for readers. Furthermore, the key objectives and methodology are clearly outlined, which aids in providing a quick overview of the study's approach. Overall, the language use in the paper is of high quality, effectively conveying the research findings. Only minor changes are required to further enhance sentence fluency and readability.

Thanks for your comment. We have now re-written sentences throughout the document for better clarity and readability.

2   While this paper undeniably offers an efficient solution for the precise detection of droplet sizes in microfluidic devices, it is indeed important to consider the practicality of these solutions in real-world scenarios. In many actual applications, droplets are often densely packed in confined spaces, making segmentation and detection significantly more challenging. It would be highly beneficial if the authors could provide deeper insights or address the applicability of their methodology in such complex situations. Elaborating on the potential limitations and discussing strategies or adaptions to handle scenarios with densely packed droplets would enhance the paper's practical value and its relevance to a broader range of real-world microfluidics applications. This additional layer of insight could make the paper even more impactful by addressing the complexities that researchers may encounter in their work.

This work aims to improve the microfluidic image processing in complex experimental configurations such as low resolution, noise or low contrast. However, the reviewer 1 raised a fair point. Even if the dripping/jetting regimes are still a massive part of the microfluidic works, more complex systems can be studied. Although, it is difficult to build a new data set for such other configurations, we extended the discussion to give some recommendations and guidelines for future works. Please find below the modifications:

'Indeed, for simple microfluidic systems, as shown in this study, deep learning methods are already apt to enhance significantly image processing. For low spatial resolution images, we recommend to use MSRN-BAM to improve the image quality before the droplet detection. Moreover, if the images shown a Gaussian noise below $\sigma\leq 4$ then DnCNN can be used along MSRN-BAM. For the droplet detection, SAM+CHT should be preferred to CHT to increase the detection rate and decrease the uncertainty on the droplet diameter, especially for low contrast images.
 Although the methodologies used here aim to be extended to more complex systems, as densely packed droplets are often discussed in the literature [38], some potential limitations can be already raised. For example, future work should focus on the segmentation and detection of such complex configurations where the droplets may overlap or be deformed by the confinement. One strategy could be to develop hybrid models that merge classical image processing techniques with advanced ML approaches. Given the added complexity where droplets overlap or deform due to confinement, a promising strategy might involve the utilisation of a specific U-Net architecture which can be tweaked to accommodate the distinction between closely packed droplets or discerning the boundaries of deformed droplets. Furthermore, active learning can be incorporated enabling the model to continuously improve its predictive performance over time. With active learning, the model could identify the most informative samples and then effectively learn to recognise other complex examples with fewer annotated examples.'

3   Figure 7 illustrates the training history of MSRN-BAM, and it would enhance the paper's clarity if the authors provided a more comprehensive explanation of the stopping criteria used during the training process. The figure suggests that the validation peak signal-to-noise ratio (Val PSNR) is still showing gradual improvement, prompting the consideration that continued training might potentially result in the development of a state-of-the-art model. A more detailed account of the rationale behind the chosen stopping point and an exploration of the possibility of further refining the model would offer readers a deeper understanding of the methodology and its potential for achieving cutting-edge performance.

As suggested by the reviewer the training process seen in Fig. 7 was detailed. Indeed, while the validation PSNR showed gradual improvement we determined that continuing training beyond 400 epochs might lead to marginal gains in performance, it would not result in significant improvements. The MSRN-BAM model was examined during training, as is common with these type of networks the performance fluctuated reaching a plateau around epoch 310. At that point the improvement in the PSNR was less than 0.03 for the remaining training time. Further training would likely result in longer training times and increased computational costs without a substantial increase in performance and may result in model overfitting. Please find below the manuscript modifications:

'Notably, the PSNR computed on the validation dataset fluctuates during training, reaching a more stable behaviour after epoch 310. The fluctuations in the PSNR during training observed in Figure 7 are common in neural network training. As such drops in the PSNR during training such as in epoch 80 for x4 scale and epoch 190 in x8 could be attributed to an update in model parameters that deviated from the optimal configuration, leading to a temporary decline in performance. This unstable behaviour during training can be attributed to several factors such as the stochastic nature of gradient descent. The model training was stopped at 400 epochs since the PSNR did not improve by more than 0.03 in the last 50 epochs.'

4    In Figure 7, a conspicuous feature is the abrupt decline in performance around epoch 190, particularly noticeable in the red line of the x8 experiment. Such drops are not uncommon in the realm of CNNs and could be attributed to factors like overfitting, variations in the training data, or fluctuations in learning rates. To enhance the figure's interpretability, a concise explanation by the authors regarding the cause of this dip would be valuable.

Indeed, fluctuations and a sharp decline in the performance during training is common with CNNs. This behaviour can be attributed to several factors as indicated. Mainly, the stochastic nature of gradient descent (in this case Adam optimizer was employed) can lead to fluctuations in performance metrics such as PSNR during training. This could be attributed to an update in model parameters that deviated from the optimal configuration, leading to a temporary decline in performance. As mentioned, the update in the model parameters can have caused the model to overfit. However, further training shows the model reaches a stable performance. The previous manuscript modifications should add more information for the readers.

Reviewer 2 Report

This study investigates the use of deep learning methods for the accurate detection and measurement of droplet diameters and the image restoration of low resolution images in microfluidics. The study uses a Phantom VEO 1310 high-speed camera with a Nivatar 24x lens to capture images of droplets generated in a microchannel. The images are then processed using the Segment Anything Model (SAM), a deep learning model that provides superior detection and reduced droplet diameter error measurement compared to the widely implemented Circular Hough transform. The study also trains a deep learning super resolution network MSRN-BAM on a dataset of droplets in a flow focusing microchannel to super resolve images for scales x2, x4, x6, x8. The super-resolved images obtain comparable detection and segmentation results to those obtained using high resolution images. Finally, the study shows the potential of deep learning in other computer vision tasks such as denoising for microfluidic imaging. I have the following comments/questions that authors may find helpful.

How does the performance of the deep learning models used in this study compare to other state-of-the-art methods for droplet detection and measurement in microfluidics?

How does the performance of the deep learning models vary with changes in experimental conditions, such as changes in droplet size, flow rate, or channel geometry?

What are the specific recommendations or future directions for research based on the findings of this study, and how might they be implemented in practice?

Author Response

We thank the reviewer for their careful reading of our manuscript and constructive feedback and respond below (in blue) to the queries raised.
The modifications made to the paper are highlighted in red to facilitate the review of our revised paper.

Comments to the Author

This study investigates the use of deep learning methods for the accurate detection and measurement of droplet diameters and the image restoration of low resolution images in microfluidics. The study uses a Phantom VEO 1310 high-speed camera with a Nivatar 24x lens to capture images of droplets generated in a microchannel. The images are then processed using the Segment Anything Model (SAM), a deep learning model that provides superior detection and reduced droplet diameter error measurement compared to the widely implemented Circular Hough transform. The study also trains a deep learning super resolution network MSRN-BAM on a dataset of droplets in a flow focusing microchannel to super resolve images for scales x2, x4, x6, x8. The super-resolved images obtain comparable detection and segmentation results to those obtained using high resolution images. Finally, the study shows the potential of deep learning in other computer vision tasks such as denoising for microfluidic imaging. I have the following comments/questions that authors may find helpful.

1    How does the performance of the deep learning models used in this study compare to other state-of-the-art methods for droplet detection and measurement in microfluidics?

 The focus of this paper is on bright-field measurements in microfluidic systems. As quickly mentioned in the introduction, for bright-field setup the main automatic method uses for the droplet detection is currently the Circle Hough Transform (CHT). But, in a lot of papers the authors just used manual measurements. However, manual measurement uncertainty depends of the operator, which is difficult to quantify. For this reason our models are only compare with the CHT method. To be clearer on the aim of this work the introduction was slightly modified to the following:
'In labs, even if many techniques were developed [5], bright-field imaging stays the most common method of monitoring droplet generation in microfluidic devices [6].'
and
'Moreover, classic droplet identification methods as manual measurements or the popular Circle Hough Transform (CHT), massively used in popular toolbox of image processing (MATLAB, Python or ImageJ), need a distinct contrast at the droplet interface to show good performances.'

2    How does the performance of the deep learning models vary with changes in experimental conditions, such as changes in droplet size, flow rate, or channel geometry?

Since the deep learning models are based on the droplet images, we expect no performance change if we change the geometry and/or the flow rates if the acquisition system is well prepared (camera exposure time, camera frequency, illumination of the channel,...). However, as seen in Fig.8 when the droplet diameter is close to the channel depth, we can observe a drop of performance for the droplet detection. The text was slightly modified to highlight this effect:

'This drop of performance could be attributed to the challenge of discerning the droplet interface from the channel wall when the droplet size is similar to the channel depth (390 $\mu$m).'

3    What are the specific recommendations or future directions for research based on the findings of this study, and how might they be implemented in practice?

Thanks to the reviewer 2 we extended the discussion to give some recommendations and guidelines for future works. Please find below the modifications:

'Indeed, for simple microfluidic systems, as shown in this study, deep learning methods are already apt to enhance significantly image processing. For low spatial resolution images, we recommend to use MSRN-BAM to improve the image quality before the droplet detection. Moreover, if the images shown a Gaussian noise below $\sigma\leq 4$ then DnCNN can be used along MSRN-BAM. For the droplet detection, SAM+CHT should be preferred to CHT to increase the detection rate and decrease the uncertainty on the droplet diameter, especially for low contrast images.
 Although the methodologies used here aim to be extended to more complex systems, as densely packed droplets are often discussed in the literature [38], some potential limitations can be already raised. For example, future work should focus on the segmentation and detection of such complex configurations where the droplets may overlap or be deformed by the confinement. One strategy could be to develop hybrid models that merge classical image processing techniques with advanced ML approaches. Given the added complexity where droplets overlap or deform due to confinement, a promising strategy might involve the utilisation of a specific U-Net architecture which can be tweaked to accommodate the distinction between closely packed droplets or discerning the boundaries of deformed droplets. Furthermore, active learning can be incorporated enabling the model to continuously improve its predictive performance over time. With active learning, the model could identify the most informative samples and then effectively learn to recognise other complex examples with fewer annotated examples.'
